# Opportunities for improved HIV prevention and treatment through budget optimization in Eswatini

Mark Minnery[1], Nokwazi Mathabela[2], Zara Shubber[3], Khanya Mabuza[4], Marelize Gorgens[3], Nejma Cheikh[3], David P. Wilson[1,5,6,7], Sherrie L. Kelly[1] *

1 Burnet Institute, Melbourne, Australia, 2 Independent, formerly National Emergency Response Council on HIV/AIDS, Mbabane, Eswatini, 3 World Bank Group, Washington, DC, United States of America, 4 National Emergency Response Council on HIV/AIDS, Mbabane, Eswatini, 5 Kirby Institute, University of New South Wales, Sydney, Australia, 6 University of Maryland, Baltimore, Maryland, United States of America, 7 Monash University, Melbourne, Australia

* sherrie.kelly@burnet.edu.au

**Data Availability Statement:** All relevant data are within the paper or detailed in the references provided in the paper, or contained in the Supporting Information files. The optima HIV

## Abstract

### Introduction

Eswatini achieved a 44% decrease in new HIV infections from 2014 to 2019 through substantial scale-up of testing and treatment. However, it still has one of the highest rates of HIV incidence in the world, with 14 infections per 1,000 adults 15–49 years estimated for 2017. The Government of Eswatini has called for an 85% reduction in new infections by 2023 over 2017 levels. To make further progress towards this target and to achieve maximum health gains, this study aims to model optimized investments of available HIV resources.

### Methods

The Optima HIV model was applied to estimate the impact of efficiency strategies to accelerate prevention of HIV infections and HIV-related deaths. We estimated the number of infections and deaths that could be prevented by optimizing HIV investments. We optimize across HIV programs, then across service delivery modalities for voluntary medical male circumcision (VMMC), HIV testing, and antiretroviral refill, as well as switching to a lower cost antiretroviral regimen.

### Findings

Under an optimized budget, prioritising HIV testing for the general population followed by key preventative interventions may result in approximately 1,000 more new infections (2% more) being averted by 2023. More infections could be averted with further optimization between service delivery modalities across the HIV cascade. Scaling-up index and self-testing could lead to 100,000 more people getting tested for HIV (25% more tests) with the same budget. By prioritizing Fast-Track, community-based, and facility-based antiretroviral refill options, an estimated 30,000 more people could receive treatment, 17% more than

model visual interface can be accessed at: http://hiv.optimamodel.com/#/login, alternately Optima HIV sourcecode can be accessed at: https://github.com/optimamodel/optima.

**Funding:** Funding for this study was provided by the World Bank Group.

**Competing interests:** The authors declare no conflict of interest.

baseline or US$5.5 million could be saved, 4% of the total budget. Finally, switching non-pregnant HIV-positive adults to a Dolutegravir-based antiretroviral therapy regimen and concentrating delivery of VMMC to existing fixed facilities over mobile clinics, US$4.5 million (7% of total budget) and US$6.6 million (10% of total budget) could be saved, respectively.

## Significance

With a relatively short five-year timeframe, even under a substantially increased and optimized budget, Eswatini is unlikely to reach their ambitious national prevention target by 2023. However, by optimizing investment of the same budget towards highly cost-effective VMMC, testing, and treatment modalities, further reductions in HIV incidence and cost savings could be realized.

## Introduction

Substantial scale-up of HIV testing and treatment in Eswatini has led to a 44% decrease in new HIV infections; from 23.1 new infections per 1,000 adults aged 15–49 years in 2014 to 15.4 in 2018 [1, 2]. This progress together with reductions in disability and HIV-related deaths are tied to continuous increases in national and international funding of the HIV response [3]. Despite encouraging progress, HIV remains Eswatini's leading cause of disability and death [4], with one the highest rates of HIV incidence in the world [1].

To end the HIV epidemic in Eswatini, onwards transmission of the virus must be prevented. In line with the 2014 call from UNAIDS for increased investment in HIV prevention to 25% of total HIV budgets [5], the Government of Eswatini made prevention one of the four key tenants of the HIV response in their 2018–2023 extended National Strategic Framework (eNSF) [6, 7]. The country's target is to reduce new HIV infections by 85% over the five-year eNSF period. The proposed strategy to achieve this reduction is to implement high-impact core interventions including a combination of HIV prevention strategies, to scale up treatment and care services, and to strengthen cross-cutting areas to produce an enabling environment for improving gender equity and empowerment of women [7].

Reaching eNSF targets in Eswatini will depend on resource availability. Despite the substantial burden of HIV in Eswatini, continued increases in HIV donor funding is not expected, rather funding cuts are anticipated in the future [3]. Moreover, stalling national economic growth and ongoing recession caused by inadequate public financing may result in decreased government spending on HIV [8]. Therefore, to maximize health gains more cost-effective investment of available HIV resources is essential.

A previous allocative efficiency modelling analysis conducted in Eswatini demonstrated that substantial gains may be possible by better allocating resources across the current mix of broad HIV program categories, such as HIV testing and treatment [9]. Recommendations from this study were considered by the national government and Ministry of Health as part of the HIV strategic planning process. In addition to allocative efficiencies, it was noted that further gains could also be realized by improving efficiencies in service delivery. In 2016, Eswatini's national AIDS program implemented a differentiated care approach for delivery and refill of antiretroviral (ARV) drugs [10]. Furthermore, in 2019, the national government highlighted the need for expansion of HIV testing service modalities to better target testing resources towards increasing HIV diagnoses [11]. This is with the recognition that a one-size-fits-all

model of HIV service delivery will not lead to the greatest success in providing sustainable and efficient service delivery. Improving implementation by optimizing funding across both broad HIV programs but also the most cost-effective and efficient prevention and care modalities may prevent more new HIV infections and HIV-related deaths with less resources, without sacrificing quality of care.

In the face of potential funding cuts, we explored approaches to improve the cost-effectiveness of HIV prevention and treatment in Eswatini through a modelling analysis in line with eNSF objectives. We focused on two areas of efficiency, allocative and implementation efficiency. To highlight potential improvements in these two areas we modelled an optimized allocation of Eswatini's most recently reported national annual HIV budget, US$123 million in 2017, of which $63 million was invested in programs whose direct impact on the HIV epidemic could be readily modelled. The impact of improving both areas of efficiency was determined by estimating the number of HIV infections and HIV-related deaths that could be averted and the amount of program coverage and cost savings that could be gained.

## Methods

### Mathematical model

Optima HIV (hiv.optimamodel.com version 2.6.11), a dynamic population-based compartmental model, was applied in this study [12]. Optima HIV has been used to conduct national and subnational HIV investment cases and strategy development in other settings. Various examples of findings from Optima HIV studies being used to help inform evidence-based HIV strategies can be found on the aforementioned Optima website. The Optima HIV model tracks HIV transmission dynamics between population groups incorporating assumptions around interactions between populations, behavioural parameters, and transmissibility of HIV. The epidemic model is overlaid with a module incorporating the cost, coverage, and outcome of programs targeted at reducing new HIV infections, HIV-related morbidity, and HIV-related deaths, such as antiretroviral therapy (ART) and condom programs. The effect of each program on the epidemic is informed by program spending, coverage, and unit costs over time. An adaptive stochastic descent optimization algorithm is applied within the model to estimate the optimized resource allocation for a given budget level against defined constraints and objective function weights. The weighting used for this analysis was 1 to 1 for minimizing new HIV infections to HIV-related deaths. The algorithm forms probabilistic assumptions about which parameters, such as changes in spending on programs that will influence prevention, treatment, and/or other outcomes, will have the greatest effect on minimizing infections and deaths [12].

The total HIV budget of US$123 million invested in 2017 in Eswatini was optimized to minimize new HIV infections and HIV-related deaths between 2018 and the end of 2022 in line with Eswatini's 2018–2023 extended National Strategic Framework (eNSF) [7]. For this analysis, an existing Optima HIV model for Eswatini from previous modeling exercises [9, 13, 14] was updated with more recent data in consultation with partners from the Eswatini Ministry of Health and the World Bank Group.

### Model inputs and calibration

The model was informed using demographic, epidemiological, and behavioural data and estimates by population group, along with programmatic expenditures and coverage levels from 2000 to 2017. Values used to inform the model, as well as their data sources, are summarized in S1–S6 Tables.

We initialized the model in 2000 using existing data to produce projections from 2019 to 2023. We calibrated the model to data and estimates for population size, HIV prevalence by population, numbers of people diagnosed with HIV, and those on ART for 2000 to 2017. Comparisons of goodness of fit to estimates from other models for new HIV infections, HIV-related deaths, and people living with HIV (PLHIV) were carried out. Calibration curves are shown in the S1–S3 Figs. Projections were then produced from 2019 to 2023.

## Cost functions

The cost-effectiveness of each HIV program or intervention in the model is defined by its cost function. Costs functions are defined by the relationship between spending and coverage and coverage and outcome, reflecting the number of individuals reached by the program for each dollar spent and the impact of this spending. Cost functions also model the increasing marginal cost associated with covering higher proportions of a targeted population. As costs increase, cost functions can become saturated, which reflects the point where increasing a program budget will no longer result in increased coverage, analogous to the challenges in providing services to the hardest to reach individuals within a population. Cost functions for each intervention can be found in supplementary material S2.

## Allocative efficiency analysis approach

The model was applied to analyse the allocative efficiency of the current HIV response using the latest reported spending by HIV program. HIV programs considered within the optimization were those that have a direct and readily measurable impact on reducing HIV transmission, morbidity, and/or mortality, including prevention and treatment programs, referred to as targeted programs. Targeted HIV programs included in this study were; efforts to keep girls in school (i.e., conditional cash transfer programs targeting adolescent girls and young women aged 15 to 24 years); condom distribution programs; social and behaviour change communication (SBCC); voluntary medical male circumcision (VMMC); antiretroviral-based prophylaxis, including pre- and post-exposure prophylaxis; HIV prevention and testing programs targeting female sex workers (FSW); HIV prevention and testing programs targeting men who have sex with men (MSM); HIV testing services (HTS); linkage to care appointment support and telephone follow-up; enhanced adherence counselling; text messaging appointment reminders; text messaging adherence support; pre-ART tracing for those who miss their appointments; prevention of mother-to-child HIV transmission (PMTCT); and antiretroviral therapy (ART). Non-targeted HIV programs, such as program management and human resources, are an important part of the HIV response, but whose effect cannot be readily measured. As such non-targeted HIV programs were not considered within the optimization but handled by fixing their spending. Certain HIV program implementation costs, such as infrastructure costs, were also classified as non-targeted and not considered within the optimization. Non-targeted HIV programs for which spending data was available, but were not considered in the optimization included programs for orphans and vulnerable children (OVC) affected by HIV; infrastructure; enabling environment; human resources; management; monitoring and evaluation (M&E); social protection; other non-disaggregated HIV prevention costs; other HIV care costs; and other HIV costs. Costing data used to inform the model were from a provider perspective as the analysis was conducted primarily to inform national strategic decision making. Out-of-pocket costs incurred by people accessing HIV services were not included.

An optimization algorithm was applied to estimate the most cost-effective allocation of resources across the above combination of HIV interventions to minimize new HIV infections and HIV-related deaths from 2019 to 2023. The latest reported HIV budget for Eswatini was

US$123 million for 2017, of which US$63 million was invested in targeted HIV programs and considered in the optimization. We also applied the algorithm to sequential increases and decreases of the most recent budget (50%, 90%, 100%, 110%, 150%, and 200%) to estimate the potential impact of varying budget on optimized allocation. HIV program spending, unit cost, and saturation values are shown in Table 1.

### Implementation efficiency analysis approach

We used the model to estimate the potential impact of four strategies to identify potential efficacies in differentiated care and service implementation.

First, the impact of switching from mainstream ART to optimized differentiated ART care modalities for stable patients (described in Table 2) was estimated. This included assessing costs, coverage, retention, and viral suppression for each modality. The potential for increased treatment coverage for people living with HIV and the number of infections and deaths that could be averted were also assessed. Improvements in efficiency based on optimized differentiated care was examined in two ways: [1] using the ART budget level from the allocative efficiency optimization and optimally redistributing this budget across the most cost-effective antiretroviral (ARV) refill options and [2] using the level of ART coverage from the allocative efficiency optimization achieved via mainstream ARV refill alone and optimally distributing surplus budget that could be gained through less expensive refill modalities across the most cost-effective refill options.

Second, we estimated an optimized resource allocation across HIV testing modalities (described in Table 3). The optimized allocation for the HIV testing budget to minimize new HIV infections and HIV-related deaths was determined as part of the overall optimization. The budget amount for HIV testing was then optimized across the most cost-effective HIV testing modalities, using unit cost per HIV diagnosis to represent cost-effectiveness of each testing modality. Ranges for maximum coverage (i.e., saturation) for each modality were agreed upon with national representatives.

Third, we explored the potential costs that could be saved by switching non-pregnant adults from the standard ART to a Dolutegravir (DTG) regimen. DTG has a 25% lower drug unit cost than the standard ART and has been shown to be safe and at least as effective as other regimens in averting HIV-related disability-adjusted life years [19].

Finally, we estimated the impact of potential efficiencies in implementation efficiency of voluntary male medical circumcision (VMMC) service delivery. Namely potential savings that could be realized if start-up costs for mobile clinic VMMC delivery were eliminated through centralization of VMMC at existing fixed healthcare sites. Efficacy for these two VMMC modalities did not differ, only unit cost.

## Results

### Potential impact of an optimized budget

In Eswatini, 85% of those diagnosed with HIV received ART in 2016–2017 [2]. Based on our optimization of the most recent budget and towards increasing treatment coverage to prevent deaths and infections through treatment as prevention, it is first recommended to increase investment in HIV testing from 3% of the total budget reported for 2017 to 6% for 2018 to the end of 2022 (Fig 1). This will allow more people to be diagnosed with HIV so they can modify their risk behaviour, prevent onwards HIV transmission, receive treatment, and achieve viral suppression. Eswatini has a generalized HIV epidemic, but there is also ongoing transmission from commercial sex in the country. As such, findings from our analysis suggest that HIV testing and prevention programs targeting female sex workers should be prioritized. In addition,

**Table 1. HIV program spending, unit cost, and saturation.**

| HIV programs and modalities | Spending (USD) | Unit cost (USD) | | | Saturation | |
|---|---|---|---|---|---|---|
| | | Low | High | Year last reported | Low | High |
| HIV prevention | | | | | | |
| Condom programs | $2,893,393 | $4.54[a] | $5.55[a] | 2016 | 75% | 80% |
| Efforts to keep girls in school | $61,083[a] | $25.00[a] | $36.00[a] | 2016 | 40% | 80% |
| PEP (combined with PrEP as ARV-based prophylaxis) | $0 | $18.00[a] | $28.00[a] | 2018 | 2% | 30% |
| PrEP (combined with PEP as ARV-based prophylaxis) | $78,840 [15] | $179.00 [15] | $219.00 [15] | 2018 | 35% | 36% |
| Programs targeting men who have sex with men (MSM) | $573,413[b] | $158.85[b] | $194.15[b] | 2017 | 70% | 80% |
| Programs targeting female sex workers (FSW) | $155,140[b] | $100.00[b] | $125.00[b] | 2016 | 80% | 90% |
| Social and behaviour change communication (SBCC) | $4,226,272 | $12.91 | $15.78 | 2016 | 80% | 90% |
| VMMC—fixed sites/public integrated | $979,810[c] | $115.00[c] | $145.00[c] | 2013 | 40% | 80% |
| VMMC—other costs | $2,217,106[c] | $433.00[c] | $529.00[c] | 2016 | 40% | 80% |
| VMMC—outreach | $3,177,533[c] | $115.00[c] | $155.00[c] | 2018 | 40% | 80% |
| HIV testing | | | | | | |
| HIV testing—overall | $4,190,502 | $10.13 | $12.39 | 2017 | 80% | 95% |
| HIV testing—home-based | $4,598[g] | $9.90[g] | $12.10[g] | 2017 | 40% | 60% |
| HIV testing—index | $14,750[f] | $4.50[f] | $5.50[f] | 2017 | 10% | 20% |
| HIV testing—mobile | $1,952,610[e] | $27.00[e] | $33.00[e] | 2017 | 55% | 65% |
| HIV testing—other provider-initiated testing and counselling (PICT)[m] | $1,604,414 | $6.30 | $7.70 | 2016 | 45% | 65% |
| HIV testing—self-testing | $56,050[h] | $22.50[h] | $27.50[h] | 2017 | 35% | 45% |
| HIV testing—voluntary counselling and testing (VCT)[n] | $684,504 | $8.10 | $9.90 | 2016 | 20% | 30% |
| Treatment | | | | | | |
| Antiretroviral therapy (ART) | $39,521,381[a] | $202.37[a] | $247.34[a] | 2017 | 90% | 99% |
| Dolutegravir (DTG)-based ART | NA | $181.00[d] | $221.23[d] | 2017 | 90% | 99% |
| Prevention of mother-to-child transmission (PMTCT) | $7,232,881[b] | $707.40[b] | $846.61[b] | 2017 | 100% | 100% |
| ART refill modality | | | | | | |
| Community-based group ART | $212,044[i] | $165.09[i] | $201.77[i] | 2017 | 20% | 45% |
| Facility-based group ART | $793,587[j] | $169.05[j] | $206.62[j] | 2017 | 30% | 50% |
| Fast-Track ART | $1,137,729[k] | $151.07[k] | $184.64[k] | 2017 | 50% | 60% |
| Mainstream ART | $12,520,055[a] | $204.87[a] | $250.40[a] | 2017 | 100% | 100% |
| Outreach ART | $461,235[l] | $409.38[l] | $500.35[l] | 2017 | 70% | 80% |

[a]Calculated based on the National Operational Plan [6],

[b]Calculated based on data from the Global Fund to Fight AIDS, Tuberculosis and Malaria (GFATM) and the President's Emergency Plan for AIDS Relief (PEPFAR),

[c]Calculated based on expert opinion from PEPFAR and on the Swaziland male circumcision strategic and operational plan for HIV prevention, 2014–2018. Swaziland Government, Ministry of Health [16],

[d]Calculated based on the National Operational Plan [6] and expert opinion from UNAIDS representatives,

[e]Calculated using data from the Determined Resilient Empowered AIDS-free Mentored Safe (DREAMS) project,

[f]Calculated using data from the CommLink project managed by PEPFAR,

[g]Calculated using data from Population Services International (PSI)

[h]Calculated using data from Médecins Sans Frontières (MSF),

[i]Calculated using data from Médecins Sans Frontières (MSF) for community-based ART refill groups for Mozambique,

[j]Calculated using data from *Max*ART [17],

[k]Calculated using expert opinion from an Optima HIV application in Malawi,

[l]Calculated using data from Moreland et al. Monitoring and Evaluation to Assess and Use Results (MEASURE) Evaluation; 2013 [18],

[m]Defined as any HIV testing modality that is recommended by healthcare providers as a standard component of care,

[n]Defined as client-initiated HIV testing delivered at a free standing health facility and through community outreach.

**Table 2. Description of antiretroviral (ARV) refill modalities [10].**

| ARV refill modalities | Overview | Number of visits per year | Priority implementation site | Benefits |
|---|---|---|---|---|
| Community-based group ART | Groups of 2–6 with people taking turns visiting the facility to collect refills on behalf of themselves and other group members | Variable, 2–4 clinic visits or 4–12 community group meetings | Where there are pre-existing networks, people are in hard-to-reach areas, among families | Increased peer support, decreased visits to the facility, reduced cost |
| Facility-based group ART | Up to 20 people meet for group counselling and collection of ARVs | 4 in total (2 ARV refill visits and group + 2 clinical consultations and group) | High-volume sites in a crowded facility, for people with constraints on availability needing early morning refill appointments, special groups | Reduced waiting time, decreased congestion, peer support |
| Fast-Track ART | No consultation, direct collection of ARV refill | 4 total (2 ARV refill visits plus 2 clinical consultations) | High-volume sites, crowded facilities, where clients have constrained working hours and need early morning refills, for special groups | Reduced waiting time, decreased congestion |
| Mainstream ART | For people who require close clinical attention and/or monitoring | Variable | All ART sites | Intense clinical services available as required |
| Outreach ART | Mobile teams from facilities offer ART services in the community | 1–12 depending on the number of outreach visits a facility can fund | Where people are in hard-to reach areas | Increased access, reduced time and cost to people living with HIV |

it is also recommended to prioritise efforts to keep girls in school, condom distribution, VMMC, and ARV-based prophylaxis including pre- and post-exposure prophylaxis. It is estimated that if the US$123 million HIV budget for Eswatini reported for 2017 is optimally reinvested from 2018 to the end of 2022, approximately 1,000 more new HIV infections (2% more) and 100 more HIV-related deaths (1% more) could be averted over this period (Fig 2). This would represent an overall reduction in new HIV-infections of 45% and 48% in HIV-related deaths from in 2022 compared with 2017 levels, falling short of the eNSF goals of 85% and 50%, respectively.

## Optimization of varying budget levels

If Eswatini's HIV budget were to be reduced by 50%, even if optimally allocated, we estimate that there could be over 170% more new infections and 40% more HIV-related deaths from

**Table 3. Description of HIV testing modalities.**

| HIV testing modality | Overview | Priority implementation site | Benefits |
|---|---|---|---|
| Home-based testing | Community-based HIV testing provided door-to-door at the homes of community members | Rural areas | Accessing first-time testers and rural populations |
| Index testing | Active and systematic HIV testing of sexual partners, biological children, injecting drug users, and associates of index cases diagnosed with HIV | Facility or community-based | Reduced cost and increased yield |
| Mobile testing | Community-based HIV testing provided through mobile units | Rural areas and workplace settings | Accessing harder to reach vulnerable populations |
| Other provider-initiated testing and counselling (PITC) | HIV test is routinely recommended and offered by healthcare providers to people attending healthcare facilities as a standard component of care | All healthcare facilities | Increased HIV testing coverage among those who seek healthcare services, relatively low cost |
| Self-testing | HIV screening test to increase awareness of HIV status resulting in benefits such as reduced risk of onward HIV transmission | Facility or community-based | Accessed by harder to reach populations such as key populations, adolescents, and men |
| Voluntary counselling and testing (VCT) | Individuals presenting for and initiating HIV testing via health facilities, free standing sites, or community outreach; also known as client-initiated HIV testing | All healthcare facilities | People can seek HIV testing if they believe to be a risk of having acquired HIV, relatively low cost |

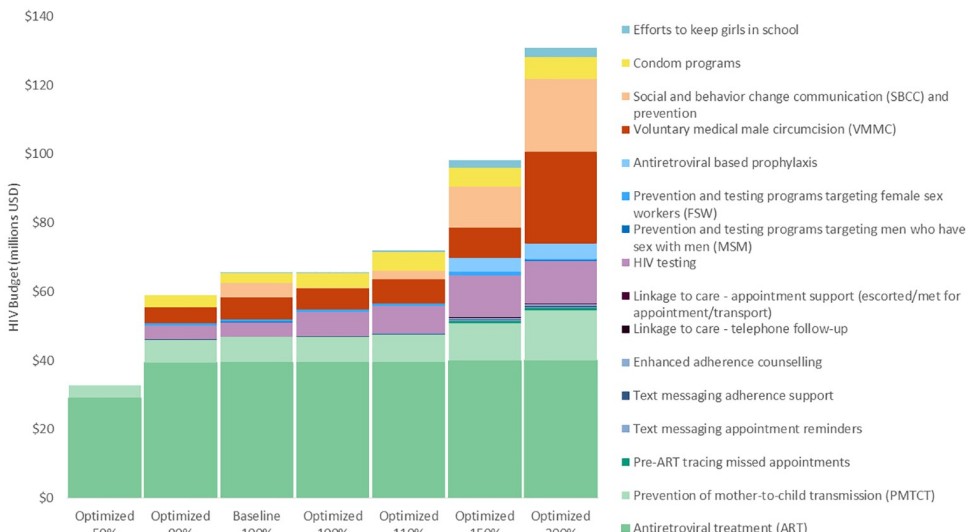

**Fig 1. Optimized allocations for varying HIV budget levels compared with baseline.**

2018 to the end of 2022 compared with baseline (Fig 2). At half the budget level under optimized allocation all programs other than treatment (including ART and PMTCT) are deprioritized. If the budget were cut by only 10% (to 90% of the latest reported amount) and optimally allocated, 1% more new HIV infections and almost 1% more HIV-related deaths could still be averted by the end of 2022 compared with maintaining the latest reported budget allocation and level.

It was estimated that if the budget level were to be increased to 110%, 150%, or 200% and optimized, an additional 7%, 15%, or 18% of new HIV infections could be averted by the end of 2022, respectively. This indicates marginal decreasing returns on investment with increasing budget somewhere above 150%. If the budget were to be increased to 110% and allocation

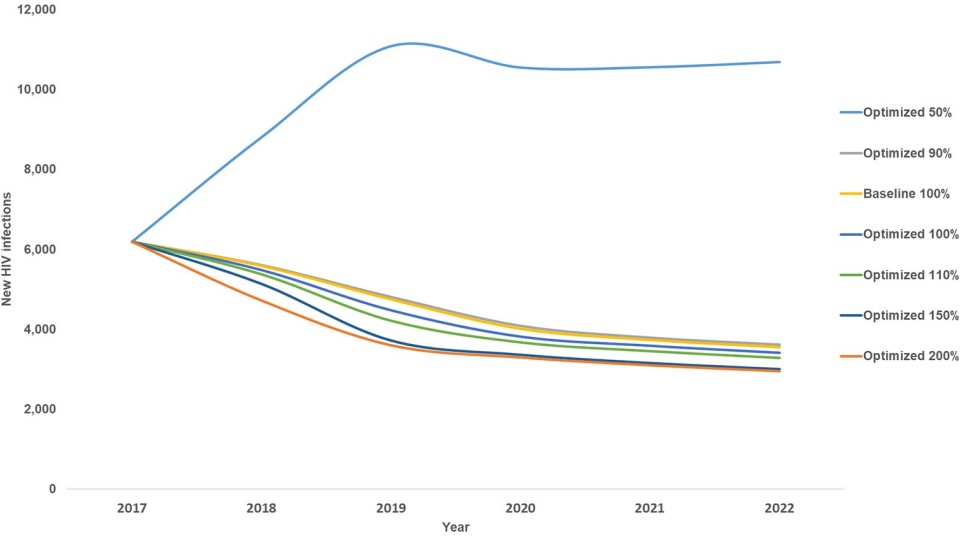

**Fig 2. Estimated new HIV infections under optimized allocations for varying HIV budget levels.**

optimized, it is recommended to prioritize HIV testing, condom programs, efforts to keep girls in school, and adherence programs. At higher optimized budget levels of 150% and 200%, after the maximum number of people are tested in a given year, further infections could be averted by prioritizing prevention programs that target the general population such as VMMC, prevention of mother-to-child transmission (PMTCT), programs to keep girls in school, ARV-based prophylaxis, social and behaviour change communication (SBCC), and to a lesser extent condom programs.

## Optimizing HIV testing and ART budgets across delivery modalities

Based on the allocative efficiency analysis, it is recommended to increase investment in HIV testing from the latest reported US$4.2 million (3% of the total HIV budget) to US$7.3 million (6% total budget) (Fig 1). We estimated that if this US$7.3 million optimized allocation was redistributed across the most cost-effective mix of HIV testing modalities, including redistributed towards index and self-testing, an additional 100,000 people could be tested by 2023, including those who are less likely to receive facility based testing (Fig 3).

We estimate that if 50% to 60% of non-pregnant adults on ART with stable viral suppression were shifted from the mainstream refill option to Fast-Track, community- and facility-based groups for their ARV refills, an additional 30,000 people could receive treatment by 2023 with the same amount of funding (Fig 4). Alternatively, if ART coverage was maintained over this period but the ART budget optimized across the most cost-effective ARV refill options, US$5.5 million in cost savings could be realized, effectively reducing the annual treatment unit cost from US$227 to US$192. These savings could then be optimally allocated following the allocations for increased budget shown in Fig 1.

We estimate that if all non-pregnant adults living with HIV were switched to a DTG-based antiretroviral regimen, $4.5 million in savings available for optimal re-investment (4% of the

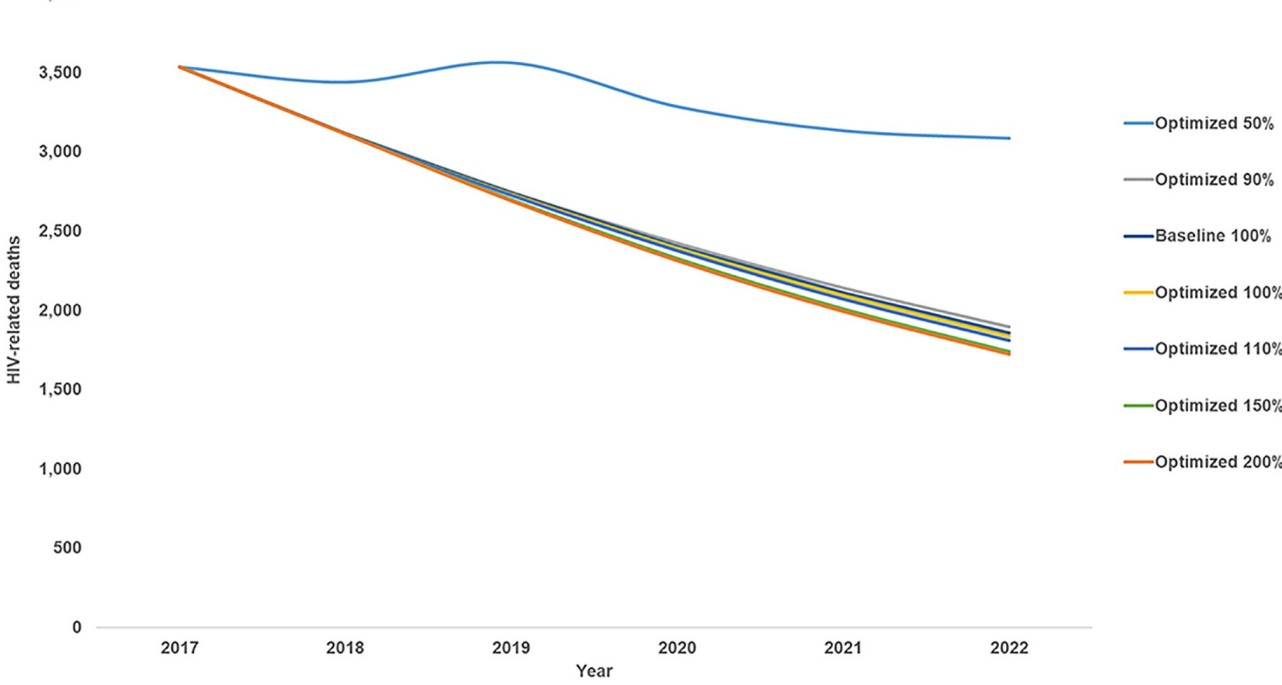

**Fig 3. Estimated HIV testing coverage under optimized allocation across testing modalities.**

(A)

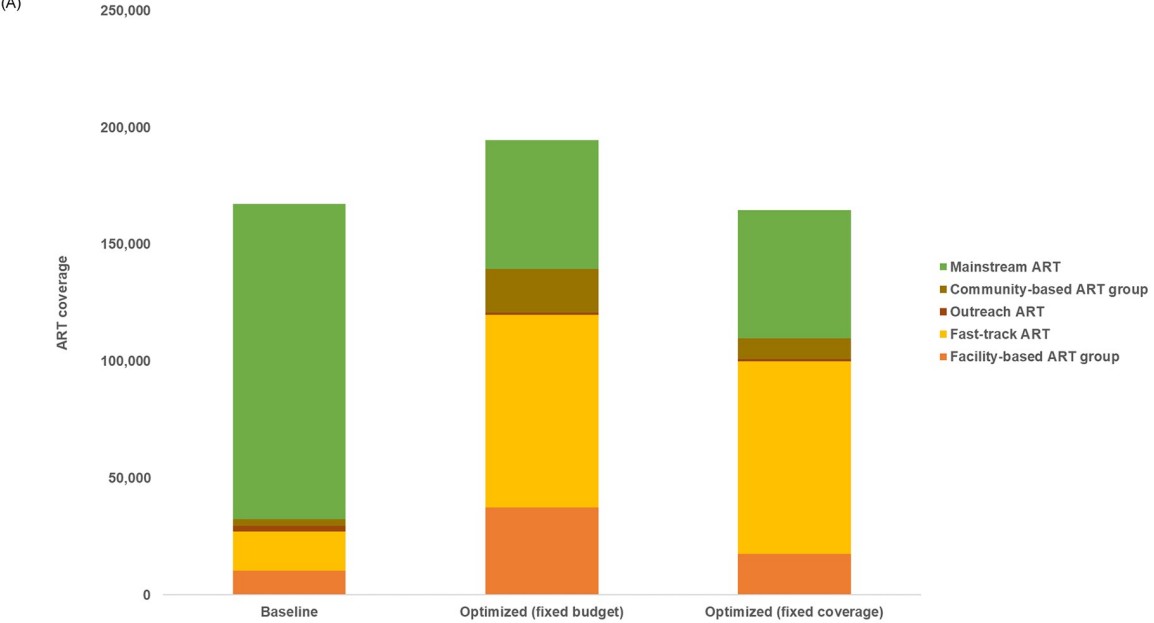

(B)

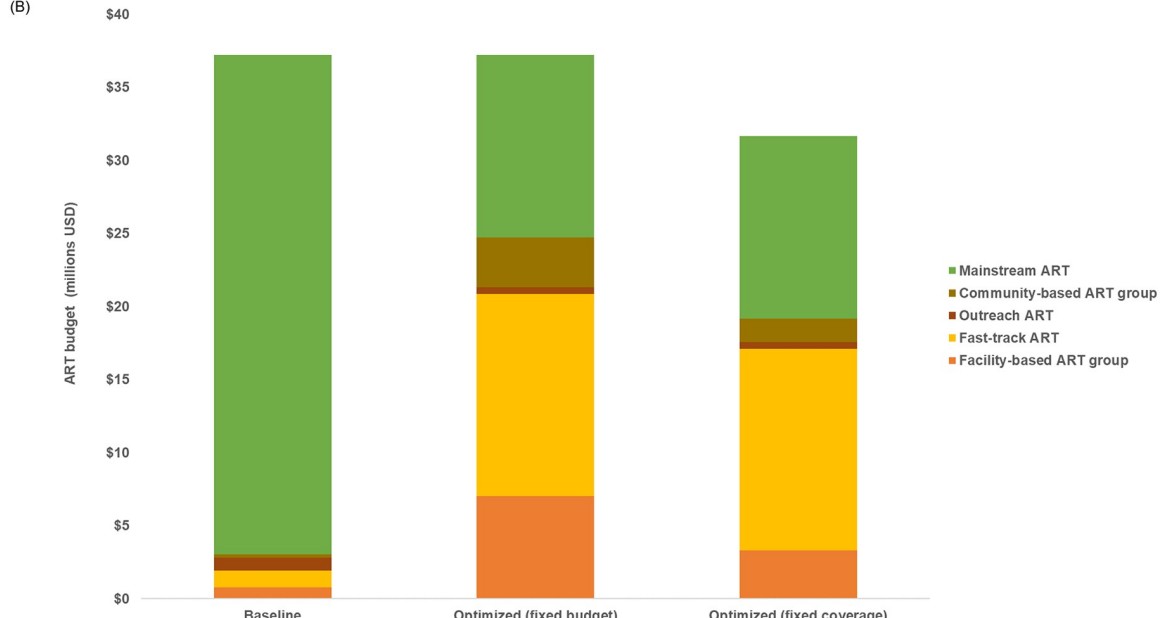

**Fig 4. Estimated budget allocation and coverage for optimization across differentiated ART care modalities.**

total HIV budget) could be realized by the end of 2022. We also estimate that by performing voluntary male medical circumcisions at existing fixed sites rather than at mobile outreach units, effectively removing start-up costs surrounding mobile units, either an annual savings of US$6.6 million could be achieved and reinvested or an additional 26,500 males could be medically circumcised each year in Eswatini. This approach to improve implementation efficiency

of VMMC could effectively result in an average program unit cost of US$140 (validated by country experts for planning purposes) compared with the latest derived unit cost of US$364 (estimated from top-down costing using data obtained from implementing partners for baseline coverage and expenditure including start-up costs).

## Discussion

Findings from this modelling study highlight the potential for greater efficiency in the HIV response in Eswatini through optimization of budget allocation and service delivery. Based on our analysis, we recommend scaling up HIV testing targeting the general population to diagnose more people who can then receive treatment. This will reduce onwards transmission of HIV; however, these results suggest a relatively modest number of infections and deaths could be averted. Recommendations provided from this analysis align with those from a previous optimization analysis conducted in Eswatini [9]. Previously it was also recommended to prioritize ART, VMMC, and HIV testing [9], which was adopted by the Government of Eswatini and the National AIDS Program. As such a large proportion of the latest reported HIV budget had already been cost-effectively committed to treatment (60% of the targeted budget), leading to 85% of those diagnosed with HIV receiving ART in 2017 [1]. With an already mainly optimized budget allocation, if recommendations from this optimization analysis were to be adopted, an additional 2% infections and 1% deaths could be averted from 2018 to the end of 2022.

Since the five-year 2018–2023 eNSF timeframe is relatively short, even under modelled optimized resource allocation it is predicted that Eswatini is unlikely to reach the eNSF prevention target of reducing new HIV infections by 85% by the end of 2022 over 2017 levels. Notably, it is not anticipated that the target would be met even if the total HIV budget were doubled and optimally allocated. The inability to reach eNSF targets even with relatively large increases in budget demonstrates the marginally decreasing return on investment in the HIV response over a short timeframe. It also highlights the effect of the inability to provide services to harder to reach populations. Achieving national prevention targets will require identifying additional cost-effective implementation efficiencies, since reaching those living with HIV who are hardest to reach will be increasingly more expensive [20]. Regardless, remarkable progress has been made by the country to reduce its incidence of HIV and HIV-related mortality. Progress could however be accelerated over the eNSF time period even with the same budget level, as demonstrated by these modeling results. If HIV funding were to be reduced as is anticipated with donor funding being withdrawn, HIV incidence can still be lessened by finding allocative and implementation efficiencies. Regardless of changes in overall budget, our analysis suggests that efficiency gains can be realized through optimizing HIV budget allocation in Eswatini, which in turn may allow increased productivity and may improve progress on human capital [21].

Our model approximates real-world constraints. These include limits on maximum attainable coverage which mimic the difficulty of rapid scale up of HIV programs. It also incorporates the increasing marginal cost associated with covering high proportions of the population. This impacts program cost-effectiveness and suggests that once a program has reached its theoretical saturation point, increased funding may be better invested in programs previously less prioritized, yet still relatively cost-effective, such as VMMC and PMTCT. Our optimized varying budget analysis demonstrates that at higher optimized budget levels (i.e., 150% and 200%), spending on ART and HIV testing make up decreasing proportions of the overall increased budget. This is due to greater estimated impact through scale-up of prevention activities to minimize HIV transmission among women, including programs which help keep girls in-

school, ARV-based prophylaxis, and HIV testing and prevention programs targeting female sex workers. This aligns with HIV epidemic trends in Eswatini where the largest proportion of HIV transmission occurs among females aged 25 to 49 years, female sex workers, and girls aged 15 to 24 years who are not attending school [11].

Should there be less funding for HIV, progress made towards Eswatini's targets could be reversed. Decreases in budget, however, remain a real possibility as donor investment in HIV has declined and is expected to continue to decline in the future [22]. With no increase in funding or reduced funding expected, now more than ever efficiencies must be leveraged and available resources optimally invested. Our findings show that implementation efficiencies can be actualized by optimizing across cost-effective modalities. For example, in Eswatini strategies for refilling ARV prescriptions may benefit from switching non-pregnant adults with stable viral suppression to less costly Fast-Track, facility- and community-based group refill options. This may in turn reduce congestion in standard refill streams and allow even more people to be sustained on treatment [10]. Community-based ART groups reduce stress on the healthcare system by allowing multiple patients to share the logistic burden of collecting their drug refills. Outreach ART allows services to be expanded and targeted towards hard-to-reach populations, and may also have additional benefits not modelled here, such as reduced out- of-pocket costs associated with collecting ARVs [10]. These recommendations, however, should be balanced with context-specific considerations not evaluated in this modelling study, such as equity of access and differences in cost associated with delivering ART in urban versus rural settings.

Efficiencies in delivering HIV testing services may be found by prioritizing index testing, which may result in higher yields. This approach increases the likelihood of diagnosing some-one living with HIV by targeting close contacts of those already diagnosed [23]. Prioritizing HIV self-testing may also lead to increased yield as hard-to-reach populations may be more accepting of this private and more accessible method. Switching non-pregnant adults on stable treatment to lower cost ARV drugs, such as DTG-based regimens, may result in cost savings. Finally, savings may be found through integration of service delivery and other approaches for health system strengthening by leveraging pre-existing infrastructure, such as delivering VMMC at fixed sites [24]. Prioritizing public fixed or integrated VMMC sites over mobile sites is likely to be more cost-effective, as it would bring about lower unit costs.

As with any modelling study, results from this analysis are estimates and should be inter-preted accordingly. Our analysis is also subject to the following limitations. First, limitations in data availability and reliability can lead to uncertainty about projected results. Contextual val-ues and expert opinion were used where available to inform the model, and otherwise evidence from systematic reviews of clinical and research studies were sourced. Although the model optimization algorithm accounts for inherent uncertainty, it might not be possible to account for all aspects of uncertainty because of poor quality or insufficient data, particularly for important cost values. Coupled with epidemic burden, cost functions are a primary factor in modelling optimized resource allocations. It is important to note that there is currently no expenditure tracking system in Eswatini and the last completed National AIDS Spending Assessment (NASA) was only available for the fiscal year 2012–2013 [25]. Therefore, expendi-tures were triangulated from different sources, applying top-down costing approaches, and are informed by expert opinion in some instances. Uncertainty bounds around estimates for new HIV-infections and HIV-related deaths are available in the supplementary information (S3 Fig). Second, we have only included costs from a provider perspective. However, as explored in this analysis we expect that ART differentiated service delivery modalities for switching peo-ple with stable viral suppression would likely result in reduced direct and indirect costs to peo-ple on care through a reduced number of clinic visits compared with mainstream ARV refill. These potential extra savings were not captured. Third, while we acknowledge the impact of

migration on the HIV epidemic in Eswatini, we did not model the potential effect migration of people living with HIV coming from countries other than Eswatini, nor the dynamics of seasonal migration. Fourth, as Optima HIV is a population-based compartmental model, the full heterogeneity for HIV acquisition risk and testing and treatment seeking behaviour may not be captured. Finally, these findings are only modelling analysis projections and have not been confirmed in a practical setting in Eswatini.

## Conclusions

Exploration of areas of efficiency within Eswatini's HIV response is crucial to further increase coverage of preventive services and reduce incidence. At latest reported budget levels, prioritizing HIV testing to diagnose more people living with HIV and to increase ART initiation is the priority for limiting onwards transmission. If additional funding can be secured, either through expanding the existing HIV budget, which is unlikely, or more likely from finding efficiency in the implementation of existing programs, optimally investing in cost-effective prevention programs will help to further reduce new infections and HIV-related deaths in the country. Finding cost-savings without sacrificing coverage or improving effectiveness of service delivery at reduced cost may be important in the face of potential austerity in HIV financing. Implementation efficiency gains across program modalities, including those for VMMC, HIV testing, and differentiated treatment care may promote an increasingly strong response to HIV into the future.

## Supporting information

**S1 Fig. Model calibration to People Living with HIV (PLHIV), new HIV diagnoses, PLHIV on treatment, new HIV infections, and HIV-related deaths.**
(DOCX)

**S2 Fig. Model calibration to HIV prevalence estimates by population.**
(DOCX)

**S3 Fig. Model calibration to new HIV infections and HIV-related deaths with projections to 2030 showing uncertainty bounds.**
(DOCX)

**S1 Table. HIV prevalence estimates.**
(DOCX)

**S2 Table. HIV testing modality coverage, yield, and mean saturation.**
(DOCX)

**S3 Table. ART refill modality coverage, efficacy, and saturation.**
(DOCX)

**S4 Table. ART refill modality coverage constraints and saturation ranges.**
(DOCX)

**S5 Table. Baseline and optimized HIV budget allocations and coverage.**
(DOCX)

**S6 Table. Optimized allocation of varying budget levels.**
(DOCX)

## Acknowledgments

The authors are grateful for the collaborative efforts of the National Emergency Response Council on HIV and AIDS (NERCHA), Eswatini Ministry of Health, UNAIDS, PEPFAR, FHI 360, PSI, CHAPS, Kwakha Indvodza, SWAGAA, SWANEPHA, CANGO, and CHAI, as well as for providing data and invaluable contextual and technical input. We acknowledge technical contributions from Rowan Martin-Hughes and model development efforts from members of the Optima Consortium for Decision Science.

## Author Contributions

**Conceptualization:** Mark Minnery, Nokwazi Mathabela, Zara Shubber, Khanya Mabuza, Marelize Gorgens, David P. Wilson, Sherrie L. Kelly.

**Data curation:** Mark Minnery, Zara Shubber, Khanya Mabuza, Nejma Cheikh, Sherrie L. Kelly.

**Formal analysis:** Mark Minnery, Zara Shubber, Sherrie L. Kelly.

**Funding acquisition:** David P. Wilson.

**Investigation:** Mark Minnery, Zara Shubber, Sherrie L. Kelly.

**Methodology:** Mark Minnery, Nokwazi Mathabela, Zara Shubber, Sherrie L. Kelly.

**Project administration:** Mark Minnery, Zara Shubber, Nejma Cheikh, Sherrie L. Kelly.

**Supervision:** Khanya Mabuza.

**Validation:** Zara Shubber, Marelize Gorgens, David P. Wilson.

**Visualization:** Mark Minnery.

**Writing – original draft:** Mark Minnery.

**Writing – review & editing:** Mark Minnery, Nokwazi Mathabela, Zara Shubber, Khanya Mabuza, Marelize Gorgens, Nejma Cheikh, David P. Wilson, Sherrie L. Kelly.

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
