## [Decision Letter · Decision Letter 0]

26 Nov 2019

PONE-D-19-25759

Opportunities for improved HIV prevention through budget optimisation in Eswatini

PLOS ONE

Dear Dr Kelly,

Thank you for submitting your manuscript to PLOS ONE. After careful consideration, we feel that it has merit but does not fully meet PLOS ONE’s publication criteria as it currently stands. Therefore, we invite you to submit a revised version of the manuscript that addresses the points raised during the review process.

We would appreciate receiving your revised manuscript by Jan 10 2020 11:59PM. To enhance the reproducibility of your results, we recommend that if applicable you deposit your laboratory protocols in protocols.io, where a protocol can be assigned its own identifier (DOI) such that it can be cited independently in the future. For instructions see: http://journals.plos.org/plosone/s/submission-guidelines#loc-laboratory-protocols

We look forward to receiving your revised manuscript.

Kind regards,

Nicky McCreesh

Academic Editor

PLOS ONE

Journal Requirements:

1. Please ensure that you refer to Figure 2 in your text as, if accepted, production will need this reference to link the reader to the figure.

Reviewers' comments:

Reviewer's Responses to Questions

**Comments to the Author**

1. Is the manuscript technically sound, and do the data support the conclusions?

Reviewer #1: Partly

Reviewer #2: Yes

2. Has the statistical analysis been performed appropriately and rigorously? 

Reviewer #1: I Don't Know

Reviewer #2: N/A

3. Have the authors made all data underlying the findings in their manuscript fully available?

Reviewer #1: Yes

Reviewer #2: Yes

4. Is the manuscript presented in an intelligible fashion and written in standard English?

Reviewer #1: Yes

Reviewer #2: Yes

5. Review Comments to the Author

Reviewer #1: Review of the manuscript “Opportunities for improved HIV prevention through budget optimisation in Eswatini”.

General comments:

The writing of the manuscript could be improved. There are sentences that are not very well structured throughout the manuscript.

I think the manuscript would be strengthened if it does not focus on opportunities alone but also includes the challenges. The title could be “Opportunities and challenges for improved HIV prevention and treatment through budget optimization in Eswatini”.

The title does not mention “treatment”, but the manuscript does include optimization of different ART care modalities. I would therefore suggest to add “treatment” to the title. See suggested title above.

In order to evaluate the validity of the study, I would like a bit more information about the Optima model and optimization functions used in the manuscript.

I am confused about the Table 1 and 2 strategies. Would it not make more sense to find the most optimal strategy among the HIV testing modalities and ART care modalities first in separate (competing choice) analyses and put the most cost-effective option in your optimization list of strategies instead of finding/optimizing the budget for e.g., treatment and then reallocating funds to the more cost-effective strategies within HIV testing modalities and ART care modalities? Currently I am not sure how you have calculated costs and effect for e.g. treatment if you do not know which ART care modality has been used (because you find the most cost-effective options after treatment budget determination).

More specific comments:

- 1st sentence Introductions: please indicate the number of new infections per year. For example: “Since 2014, substantial scale-up of HIV testing and treatment in Eswatini has led to a 44% decrease in new infections from xxxx/1000 adults to xxxx/1000 adults annually.

- The last sentence of the 1st paragraph of the Introduction section should be rephrased, because the sentence is not correct as it is.

- 2nd sentence 2nd paragraph of the Introduction: I would mention that it concerns the HIV budget by rephrasing the sentence to “In line with ….. 25% of the total HIV budget on prevention, the Eswatini government …….”.

- I would clarify that the NSF period is 5 years in the last sentence of the 2nd paragraph of the Introduction: “The proposed target was a 85% reduction in new infections by the end of the 5-year NSF period (2018-2023).”

- Can the authors add to the manuscript how the current NSF is aiming/ proposes to accomplish a 85% reduction in new HIV infections.

- I would be interested in what the current HIV budget is in Eswatini. Can the authors mention that in the Introduction?

- The authors mention that allocative efficiency modeling has taken place already, but it was unclear if this type of analysis was integrated in the 2018-2023 NSF. Can the authors make this more explicit in the text?

- It was unclear whether the analysis of the current study serves to inform a strategic document. The NSF has already been published and only ends in 2023. Can the authors clarify this? It seems like a redundant analysis if the NSF 2018-2023 has already been informed by an allocative efficiency analysis. In other words, please make clear how this analysis fits in/ add to the analysis already conducted and for which strategic document (if any) this analysis has been done.

- The authors state “Recommendations from these studies have been adopted by the country Government, and the current response may already be highly allocatively efficient.” This statement is rather vague. Do we not know? I would be interested in the degree to which Eswatini has implemented the results of this allocative efficiency analysis that has been undertaken. Can the authors mention anything about that in the text?

Methods section

- What is the budget that the authors are working with in their optimization analysis?

- Can the authors elaborate a bit more (in a couple of sentences) on the Optima HIV model for the readers who do not know the model. The information currently provided is very scarce. I am not sure what the authors mean with a compartmental model. Also, it would be good to mention that the Optima model is a commonly-used model in costing and optimization exercises for strategic documents with some references of analyses in which the Optima model has been used.

- Can the authors be more elaborate in the data sources section. The paragraph before that states that a prior model analysis has been updated. In the data sources section, I would be interested in what data has been updated and how for the new model.

- In the model calibration section, it was stated that estimates were calibrated to estimates from other models, but no references were provided here. To what other models were the estimates calibrated to?

- In the analysis approach section: how were cost functions updated? And from what? From the optimization algorithm described elsewhere? Can you add a couple of sentences describing the methodology behind the algorithm? Its hard to evaluate the validity of a study when the most crucial elements such as the optimization algorithm is not described in the current manuscript.

- In the “impact of optimized allocative efficiency” section, please mention what the currently available budget is.

- Is is unclear whether implementation costs are also considered in the program costs.

- The two strategies mentioned just before the Results section (a regimen of DTG for non-pregnant adults and VMMCs) came as a surprise. If these are considered strategies in the optimization exercise these should be mentioned in the table of available treatment and prevention strategies. Now it comes across as just trying out some things.

Results section

- It is unclear whether the strategies listed at the end of the 1st paragraph of the results section are just included in the optimization exercise or actually the chosen programs to fund based on the budget. A clarification would help. As well as a table with all strategies/programs considered for funding.

- 2nd paragraph of the results section: please specify “a large increase in infections and deaths” by mentioning the actual number of infections and number of deaths for the budget cut of 50% or the % of increase for these 2 outcomes.

- The results section mentions: “At higher optimised budget levels of 150% and 200%, after the maximum number of people are tested in a given year, further infections may be averted by prioritizing primary prevention programs that target the general population such as VMMC, prevention of mother-to-child transmission (PMTCT), programs to keep girls in school, ARV-based prophylaxis, social and behaviour change communication (SBCC), and to a lesser extent condom programs.”

What do you mean with “may”? Are you guessing or are these the programs that should be funded based on the optimization exercise?

- Page 10, last paragraph: do you mean 2023 instead of 2022?

- The Results section states: “Due to a lower unit cost, public fixed or integrated VMMC sites (over mobile VMMC sites) are likely more efficient.” If this is not a result of the analysis, mention it in the Discussion section instead.

- The 2 unit costs mentioned in the last paragraph either need to be accompanied by a source reference or an explanation of how the unit cost was estimated and what exact components it consists of.

Discussion section

- 1st paragraph: “HIV testing amongst the general population should be scaled up to diagnose more people ….”. How much more? How much scale up? Be more precise here. What type of testing you are talking about (from Table 1)?

- Have you taken into account that Eswatini has started implementing prior optimization results already? If so, how?

- I am confused (2nd paragraph): are we not talking about the 2018-2023 NSF timeframe mentioned earlier?

- They won’t make the projected 85% reduction. Mention what they make instead in the 2nd paragraph.

- “Exploring the possibility for switching non-pregnant adults on stable treatment to lower cost ARV drugs, such as Dolutegravir-based regimens, may result in cost savings”. What do you mean with “may”? You have investigated that in this study and should now know the answer.

- “Although the model optimisation algorithm accounts for inherent uncertainty, it might not be possible to account for all aspects of uncertainty because of poor quality or insufficient data, particularly for important cost values.” What do the authors mean with inherent uncertainty? I am missing sensitivity analyses for the most important cost values. This seems important as the quality of the cost data used is difficult to evaluate.

- In the conclusion section I would not mention the “If additional funding can be secured …” as that is an unrealistic scenario in the light of budget cuts. I would focus on the scenario of budget reduction and the optimal spending of the current budget.

Reviewer #2: Reviewer:

Overall, I think this is an interesting and important topic. It addressed an urgent issue in the HIV prevention in Eswatini. The findings could be used to inform and improve the current HIV practice. I have some specific suggestions below. Some expert opinions are needed for the modeling. I assumed that the modeling part was conducted properly for analyzing the questions in this paper.

Major points:

1. For Table 1, please provide what was the targeted population in each priority implementation site. For Table 1 and Table 2, please provide what the current percent coverage was for each HIV testing modality and each ART modality in the overall population.

2. On page 8, “measured the effect that switching eligible people to those modalities had on cost, coverage, retention, and viral suppression”. Were the implementation/switching costs considered or included in the analyses besides the program costs? Please explain this point in the paper.

3. On page 10, “A budget incorporating VMMC with reduced start-up costs and other implementation efficiencies results in an average unit cost of $140 (estimated unit cost cited by the country for planning purposes and other bottom-up costing analyses) compared to $364 at baseline (estimated using a top-down costing approach using data for baseline coverage and expenditure including start-up costs obtained from implementing partners).”

Are the unit costs of two scenarios comparable by using two different methods? Please justify this.

4. In the discussion session (page 12), “Our findings show that implementation efficiencies can be actualized by optimising across cost-effective modalities.”

What is the feasibility consideration regarding increasing/decreasing funding for different testing and ART modalities? How will the changes impact different groups of population, such as rural population vs. urban population, etc.? Please provider some discussion on this.

5. In the discussion session (page 12), “Community-based ART groups reduce stress on both the healthcare system and on people needing treatment refills by allowing multiple patients to share the logistic burden of collecting their drug refills.”

How would this change impact adherence? Are there studies providing evidence on this?

6. Authors mentioned several limitations in the discussion session. One of them is that “Although the model optimisation algorithm accounts for inherent uncertainty, it might not be possible to account for all aspects of uncertainty because of poor quality or insufficient data, particularly for important cost values.”

To what extent, the findings would still hold under this limitation. Have you done sensitivity analysis to explore some of those data uncertainties?

7. In the discussion, authors mentioned the study was from a provider perspective. Please mention this in the method section and justify or explain the reasons for pick this perspective.

8. For figure 2, except scenario “Optimized 50%”, other scenarios had similar trends and results regarding the outcome, new HIV infections. It would be helpful to inform how new HIV infections change in scenarios when Optimized between 50% and 90%. This would help to show what the critical points/cutoffs are for the trends and results to change.

Minor points:

9. On page 9, “If the budget were cut by 10% (to 90% of the latest reported amount), but with optimised allocation, an estimated 300 more new HIV infections (1% more) and 100 more HIV-related deaths (less than 1% more) could be averted by 2023. Optimised allocation of budgets increased to 110%, 150% and 200% are estimated to result in 7%, 15% and 18% fewer total new infections by 2023, respectively, showing that return on investment marginally decreases with increasing budget at these levels. With an increase to 110% optimised budget, HIV testing should be prioritized.”

In the first sentence, both the number of cases and the percent increase were mentioned for both HIV-related deaths and new HIV infections. In the second sentence, only the percent decrease in new infections were mentioned. Please be consistent and add in number of cases increased/decreased for both outcomes in the second sentence.

10. For figure 3, besides the number of people in the y-axis, please also provide the coverage in percent of total population.

11. Please be clear about what saturation means in the supporting information. It is not completely clear to the general audience for this journal.

6. PLOS authors have the option to publish the peer review history of their article (what does this mean?). If published, this will include your full peer review and any attached files.

Reviewer #1: No

Reviewer #2: No

---

## [Author Response · Author response to Decision Letter 0]

17 Apr 2020

Please find a comprehensive response to all reviewer comments in the attached file 'Eswatini_reviewer response.docx'

---

## [Decision Letter · Decision Letter 1]

5 May 2020

PONE-D-19-25759R1

Opportunities for improved HIV prevention and treatment through budget optimisation in Eswatini

PLOS ONE

Dear Dr Kelly,

Thank you for submitting your manuscript to PLOS ONE. After careful consideration, we feel that it has merit but does not fully meet PLOS ONE’s publication criteria as it currently stands. Therefore, we invite you to submit a revised version of the manuscript that addresses the points raised during the review process.

We would appreciate receiving your revised manuscript by Jun 19 2020 11:59PM. To enhance the reproducibility of your results, we recommend that if applicable you deposit your laboratory protocols in protocols.io, where a protocol can be assigned its own identifier (DOI) such that it can be cited independently in the future. For instructions see: http://journals.plos.org/plosone/s/submission-guidelines#loc-laboratory-protocols

We look forward to receiving your revised manuscript.

Kind regards,

Nicky McCreesh

Academic Editor

PLOS ONE

Reviewers' comments:

Reviewer's Responses to Questions

**Comments to the Author**

1. If the authors have adequately addressed your comments raised in a previous round of review and you feel that this manuscript is now acceptable for publication, you may indicate that here to bypass the “Comments to the Author” section, enter your conflict of interest statement in the “Confidential to Editor” section, and submit your "Accept" recommendation.

Reviewer #1: (No Response)

2. Is the manuscript technically sound, and do the data support the conclusions?

Reviewer #1: Partly

3. Has the statistical analysis been performed appropriately and rigorously? 

Reviewer #1: N/A

4. Have the authors made all data underlying the findings in their manuscript fully available?

Reviewer #1: Yes

5. Is the manuscript presented in an intelligible fashion and written in standard English?

Reviewer #1: Yes

6. Review Comments to the Author

Reviewer #1: Thank you for letting me review the revised version of your manuscript. I think it has improved a lot compared to the previous version and now includes a lot of information that was missing. Please find my additional concerns and comments below.

Major comment:

Were there any challenges in obtaining the data needed to populate the model? The major problem with this type of analyses is that high-quality unit cost data is often not available. Because of this challenge, comparison of actual trends in HIV/AIDS spending and program output against resource needs estimates that had been calculated for strategic planning reveal large variances. That variation is problematic in planning. The authors mention the quality of the cost data in the limitations, but as a reader (who appraises the validity of the study findings while reading) I would like to see in the Methods section what the sources of the cost data were (e.g., expert opinion, published literature, expenditure data) and how uncertain these unit cost estimates are. If there is a lot of uncertainty in these unit cost estimates, the authors should conduct sensitivity analyses on the model input parameters to investigate the impact on the cost-effectiveness results.

Minor comments:

Abstract

The introduction seems a bit long for the introduction of an abstract. Can the authors shorten this paragraph by leaving some of the details out? For example, the age range of adults can be removed (“, with 14 infections/1000 adults in 2017.”). Remove the “over 2017.” in the next sentence.

Also, please rephrase the last sentence of the introduction so it reads as the objective of the study. The aim/objective of the study is ….

Methods

The second sentence is too long. Please cut this sentence in (at least) 2 sentences.

Findings

I would mention first what changes have to be made in the adoption/funding of HIV programs when we allocate based on cost-effectiveness. Next, mention the consequences that those changes have in the prevention of infections and deaths.

Significance

Add “relatively” before “short five-year timeframe”.

A budget can either be optimally spend or not optimally spend. I don’t think there can be an increase in an optimized budget. Please consider removing “increased” in this sentence.

The government is aiming for a 85% decrease in new infections and optimizing the current budget will result in a 2% reduction? There is a huge gap there. I would love to see what the authors think the implications of these findings are. Is the 85% completely unreasonable or are there additional ways to get to a higher percentage of reductions? For example, increasing the budget to what amount will approach the 85% or is 85% completely unreasonable? Mention what a reasonable increase in budget would do (and even then we would not come close to 85%).

Introduction

Please read through one more time to improve some of the English language (e.g., 3rd paragraph: change sentence in “Reaching these HIV reduction targets in E. will be dependent on the availability of resources.” Etc.).

Methods

The authors state “Drawing from a previous modelling exercise (13), the national model for Eswatini was updated in consultation with partners from the Eswatini Ministry of Health and the World Bank Group.”. Can the authors be a bit more specific about how the current study differs from the 2018 Lancet analysis (i.e., what was updated)?

What do the authors mean with we initialized the model in 2000 in the sentence “We initialized the model in 2000 and produced projections from 2019 to 2023.”? Please clarify. Did you start using the model in 2000 or did you run it from 2000 until 2023 and using the 2019-2023 period for the current analysis?

Results

The figures could use some footnotes to remind the reader about what the different scenarios represent.

7. PLOS authors have the option to publish the peer review history of their article (what does this mean?). If published, this will include your full peer review and any attached files.

Reviewer #1: No

---

## [Author Response · Author response to Decision Letter 1]

19 Jun 2020

Please find a full itemized response to reviewer comments in the attached document: 'Eswatini HIV second response to reviewers.docx'

---

## [Editor Report · Decision Letter 2]

22 Jun 2020

Opportunities for improved HIV prevention and treatment through budget optimisation in Eswatini

PONE-D-19-25759R2

Dear Dr. Kelly,

We’re pleased to inform you that your manuscript has been judged scientifically suitable for publication and will be formally accepted for publication once it meets all outstanding technical requirements.

Kind regards,

Nicky McCreesh

Academic Editor

PLOS ONE
---

## [Editor Report · Acceptance letter]

8 Jul 2020

PONE-D-19-25759R2 

Opportunities for improved HIV prevention and treatment through budget optimization in Eswatini 

Dear Dr. Kelly:

I'm pleased to inform you that your manuscript has been deemed suitable for publication in PLOS ONE. Congratulations! Your manuscript is now with our production department. 

Kind regards, 

on behalf of

Dr. Nicky McCreesh 

Academic Editor

PLOS ONE